# Outdoor Human Thermal Comfort along Bike Paths in Balneário Camboriú/SC, Brazil

**Luana Writzl** [1], **Cassio Arthur Wollmann** [1,*], **Iago Turba Costa** [1], **João Paulo Assis Gobo** [2], **Salman Shooshtarian** [3] **and Andreas Matzarakis** [4,5]

1 Department of Geosciences, Natural and Exact Sciences Center, Federal University of Santa Maria, Santa Maria 97105-900, Brazil
2 Department of Geography, Core of Exact Earth Sciences, Federal University of Rondônia, Porto Velho 76801-059, Brazil
3 School of Property, Construction and Project Management, RMIT University, Melbourne, VIC 3001, Australia
4 Research Centre Human Biometeorology, German Meteorological Service, Stefan-Meier-Str. 4, 79104 Freiburg, Germany
5 Chair of Environmental Meteorology, Faculty of Environment and Natural Resources, Albert-Ludwigs-University, 79085 Freiburg, Germany
* Correspondence: cassio@ufsm.br

**Abstract:** This research is concerned with understanding the degree of human thermal (dis)comfort in connection with the various microclimates present in the vicinity of bike trails in Balneário Camboriú/SC, Brazil, throughout the summer. Local Climate Zones were determined using the Sky View Factor and were identified along research routes and schedules at 9:00 a.m. and 4:00 p.m. on a subtropical summer day (14 January 2022). Data were collected with weather devices attached to the bicycle, measuring air temperature, relative humidity, and globe temperature, from which the mean radiant temperature was calculated. The PET and UTCI indices were used to assess outdoor thermal comfort in the summer. The findings revealed that at 9:00 a.m., the eastern half of the city had a higher tendency for thermal discomfort; however, at 4:00 p.m., this same location had thermal comfort for users along bike routes. At 4:00 p.m., the PET index indicated that 24% of the bike paths were pleasant, and the UTCI index indicated that 100% of them were in thermal comfort. At 9:00 a.m., the majority of the city was under discomfort conditions. The index values reflect the morning time, and the study shows that there is now a negative correlation between the SVF and the indexes, which means that the greater the SVF computations, the lower the values are. The PET and UTCI indices revealed a positive association in the afternoon period: The greater the SVF values, the higher the PET and UTCI indices. Further research should be conducted in the future because many parameters, such as construction, position, and urban (im)permeability, as well as sea breeze and solar radiation, can have a significant impact on outdoor human thermal comfort in Balneário Camboriú, and when combined with the type of LCZ and the SVF, it is possible to understand how all of these active systems interact and form microclimates that are beneficial to bike path users.

**Keywords:** Balneário Camboriú; microclimate; bicycle; bike paths; mobile measurements; outdoor human thermal comfort; PET; UTCI; summer

## 1. Introduction

The growth of urban centers generates a greater interest in using outdoor spaces, making discussions about external human thermal comfort more pertinent [1]. Due to the complex interactions between urban form and function and the atmosphere, urban morphology can contribute to thermal stress in cities [2,3]. There is higher variation in microclimates close to the earth's surface, where radioactive, thermal, humidity, and aerodynamic climatic qualities combine with the physical structure and anthropogenic relationships of the urbanization present [4].

Buildings cause the most thermal interference in the external environment, which leads to problems with cities' aerodynamics. The construction material, color, texture, and surface influence the reflectivity of solar radiation (Sr) and shading, which can change the temperature due to the reduced sky view factor, as well as changes in humidity. In coastal cities, structures are significant impediments to natural ventilation from sea breezes; such effects rely on the orientation of the urban fabric, layout, height, and width of the buildings [4,5].

Furthermore, the road network has an impact on environmental factors. As a result of the high volume of vehicle traffic generating a substantial amount of anthropogenic heat emission, in turn, impacting the concentration of pollutants, the water balance surrounding the buildings is disrupted, owing to geographical variability in the reception rainfall, soil drainage, and evaporation [3–6]. Plant transpiration eliminates heat and lowers the quantity of radiation received by the surface in places with permeable cover. The presence of water bodies in a city reduces thermal stress up to 20 m from the water's edge by decreasing Ta and increasing humidity due to evaporation of surface water [3,7].

In the majority of Brazilian cities, people engage in outdoor activities, such as shopping, commuting to work, and engaging in physical activities, in severe microclimatic conditions that are unknown to science. It should be noted that when the human body does not achieve thermal neutrality appropriate to the microclimatic conditions of the environment, this can result in a variety of physical and emotional damages, as well as uncomfortable sensations, which can ruin or alter the experience of people participating in physical activities, and even in death [8–11]. Numerous studies have been conducted to measure Human Thermal Comfort in metropolitan situations, such as in German [12,13], Chinese [14–16], and Brazilian [17,18] cities.

Outdoor activities, when voluntary, experience a significantly lower demand on days with extremely high temperatures. Furthermore, individuals prefer more appealing public locations, such as gentle landscapes shaded by trees and having water resources, while they avoid less attractive landscapes [19]. It might be claimed that people are comfortable when they do not need to make adjustments to their environments [5]. In this respect, there is no "perfect comfort zone" that covers all persons in the same space/time, as individuals experience biological variances and distinct psychological processes.

In order to comprehend the mutual interactions between living organisms and the atmospheric environment, the field of human bioclimatology has developed a combination of sensory studies and detailed quantitative knowledge in an effort to find optimal environmental conditions for the greatest number of coevolving individuals; this is known as climatic variability [20]. When evaluating thermal comfort in outdoor activities, physical, physiological, psychological, and social/behavioral factors should be considered, as the state of mind can also influence the impression of thermal comfort [21,22].

There are six essential elements for measuring the exterior thermal comfort of humans. The first four are physical factors that comprise the thermal environment; they are Ta, mean radiant temperature (MRT), wind speed (Ws), and ambient water vapour pressure. In addition, activity level (internal body heat production) and thermal resistance of clothing [3,9,20,21] must be considered.

Innovative approaches have introduced the bicycle as a method for mobile measurements of environmental variables in various contexts, including research involving temperature and thermal comfort in open areas [23]. The Physiologically Equivalent Temperature (PET) and the Universal Thermal Climate Index (UTCI) are the best thermal comfort indices because they provide a complete description of the biophysiological processes that maintain the body's thermal stature.

Using the Local Climatic Zones (LCZ) and the Sky View Factor (SVF) as urban morphology descriptors to identify probable correlations between environmental variables and distinct microclimates has proven effective in studies [15,24–29]. The LCZs take into account various factors and elements of the urban environment, such as the sky view factor, the average between the height, width, and spacing of buildings and trees, the portion of

the building's surface, impermeability, permeability, the height of buildings or trees, and the terrain's roughness, among others. In this regard, it is clear that in complicated and extremely vertical urban landscapes in which environmental variables change unexpectedly, such as in Balneário Camboriú, research pertaining to outdoor human thermal comfort are crucial, particularly in connection to outdoor activities, such as in bike paths [30–33].

In Balneário Camboriú, the high volume of vehicle and pedestrian traffic in such a short and vertical space, such as urban canyons, makes it impossible for the wind to circulate freely and can generate a large accumulation of anthropogenic heat, significantly interfering with Ta, which can be negative in a tourist city. It is evident that over the years, Balneário Camboriú has increased the number of spaces for cyclists, encouraging the use of bicycles as a means of transport, which is efficient, economical, sustainable, and it also improves the quality of life for the population. In addition, understanding the interrelationships of the climatic environment enables the development of precise strategies for climate adaptation. In this regard, the research aims to comprehend the level of human thermal (dis)comfort in connection with the various microclimates resulting from the urban morphology surrounding the bicycle trails in Balneário Camboriú during the summer. To do this, we endeavored to characterize the urban morphology of the research area, making use of mobile measurements with bicycles in Brazil to assess the thermal (dis)comfort of humans.

## 2. Materials and Methods

### 2.1. Study Location

This study was conducted at Balneário Camboriú, which is located in the Southern Brazilian state of Santa Catarina and is part of the Subtropical Climate (Cfa) that runs in the state's coastal communities [34]. Its latitude and longitude are $26°59'42''$ S and $48°37'46''$ W, respectively. The Balneário Camboriú is situated on the northern shore of Santa Catarina, bordering the towns of Itajaí, Itapema, and Camboriú (Figure 1).

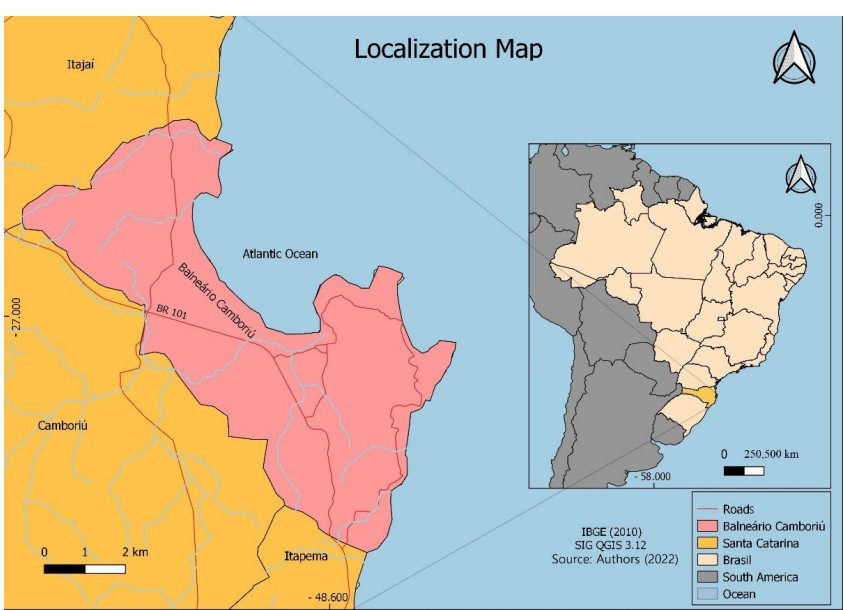

**Figure 1.** Location map of Balneário Camboriú/SC, Brazil.

According to the Brazilian Institute of Geography and Statistics (IBGE), the city has a population of approximately 146,000 people, which reaches one of the highest demographic densities in Brazil during the hot seasons, with more than one million tourists from all over the world [35]. All of this human movement occurs over an area of 45.21 km$^2$ in an environment characterized by a complex verticalization in the real estate industry. Balneário Camboriú is known as the "Brazilian Dubai" due to the presence of eight of the country's

ten tallest structures, as well as the fact that the city's verticalization process is the most intense in the entire Southern Hemisphere (Figure 2).

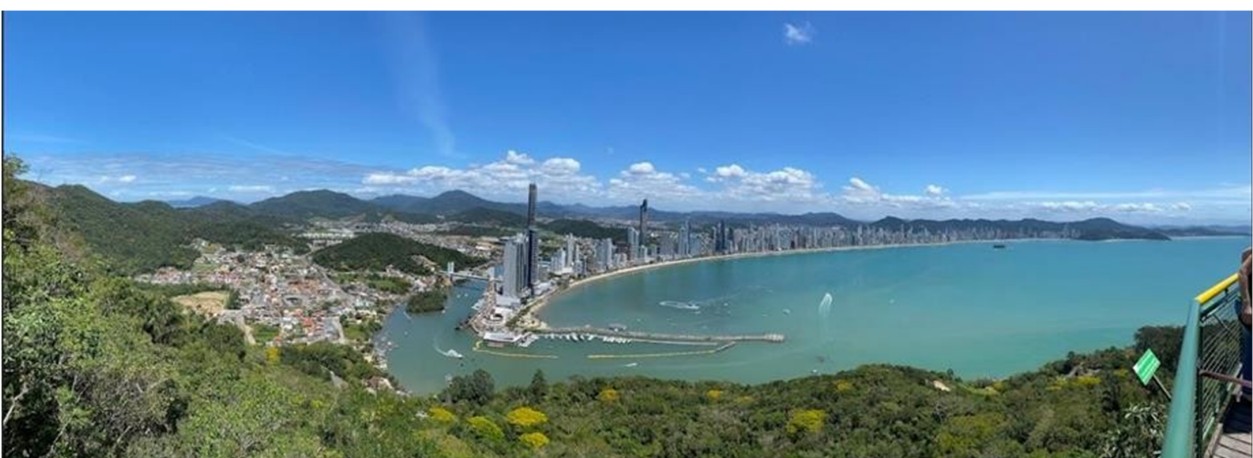

**Figure 2.** Panoramic view of Balneário Camboriú/SC, Brazil. Source: Authors, taken during field-work, December 2020.

The concentration of urban canyons is shown in Figure 2 along a 6 km stretch of shoreline [36,37]. Both the location's geomorphological structure and its limited territory make the development of skyscrapers both possible and necessary due to the region's geographical configuration [36,38].

### 2.2. Characteristics of Urban Morphology

To characterize the urban morphology of Balneário Camboriú, the Local Climatic Zones (LCZ) approach was employed. The mapping of LCZs for the city of Balneário Camboriú was obtained using the World Urban Database and Access (WUADAPT) in an effort to facilitate the process of identifying LCZs and enable a higher number of studies [39,40].

In addition, the Sky View Factor (SVF) approach was employed, which is commonly used to characterize the geometry of urban canyons by measuring the locations where the sky becomes blocked, typically by buildings or trees [23]. Their computations range from 0 to 1, with 0 indicating that the sky is entirely obscured by topography or barriers and 1 indicating a clear sky [41].

For the SVF computations, using the RayMan PRO 2010 Free software (Freiburg, Germany) [42], we employed a 218° fisheye lens connected to the camera of a cell phone (Figure 3) [5,41–47]. The places chosen for the removal of pictures varied on the basis of the results obtained from the transects' change of environment zones (LCZs) [31].

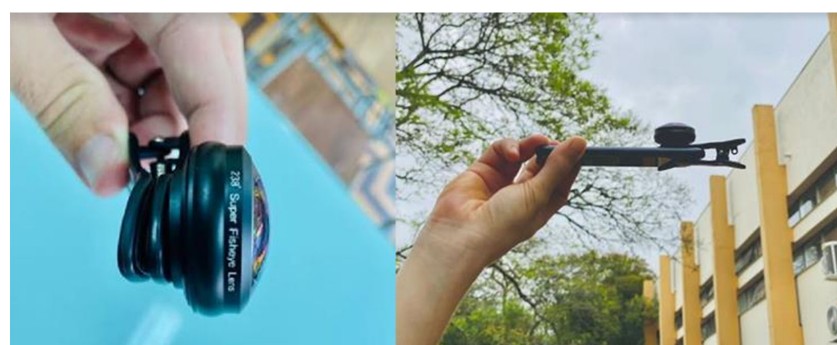

**Figure 3.** Instruments used to take the SVF photos. (Source: Photographs taken during field work). Authors (2022).

### 2.3. Mobile Measurements

Data collection, in the form of mobile measurement at the same time, took place on 14 January 2022. The measurements covered two different routes available for bicycle use (Figure 4).

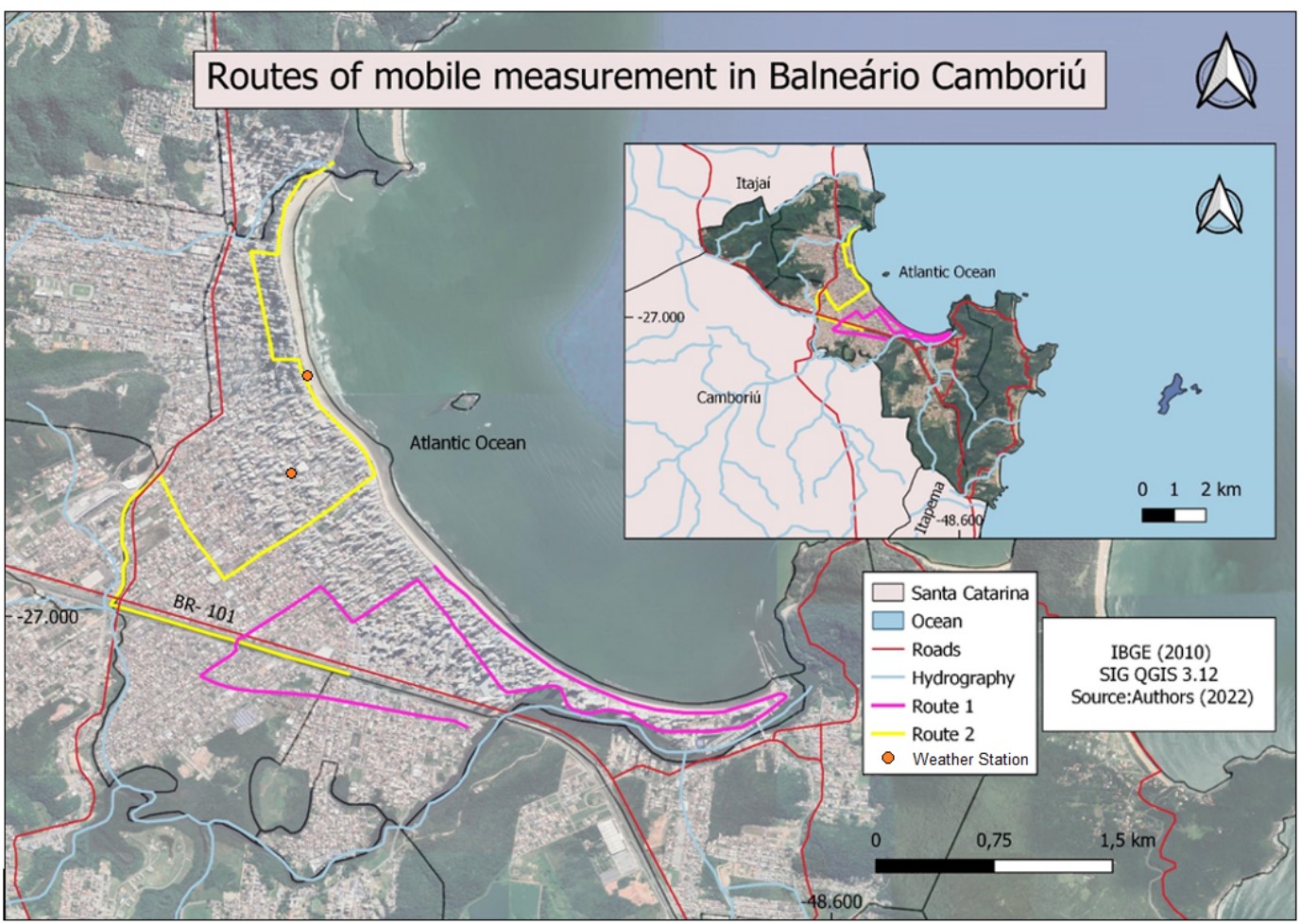

**Figure 4.** Map of mobile transects carried out in the urban area of Balneário Camboriú/SC, on 14 January 2022, at 09:00 a.m. and 4:00 p.m. Source: Authors (2022).

The times selected for the mobile measurements were at 9:00 a.m. and 4:00 p.m.; these times were defined because they have different influence on environmental variables, mainly as a function of Sr [23,32].

The bicycles used in the mobile transects were loaned by supporters of the research, living in the city of Balneário Camboriú, who became aware of the research through the disclosures that were made by the city's media. For mobile measurements, two dataloggers of the HOBO® U23-001 model were used (Figure 5). One was used to carry out measurements of Ta and Rh, and the other was used to later calculate the MRT through the Ta data obtained by the globe thermometer (Tg).

After proper calibration [36], the dataloggers were attached to the bicycle using a PVC support, (Figure 6). The datalogger that provided Ta and Rh data was protected by a smaller white PVC pipe, which was manually drilled to allow ventilation and avoid direct sunlight. It was mounted 1.5 meters off the ground, to avoid unwanted influences and to respect suggested standards [48].

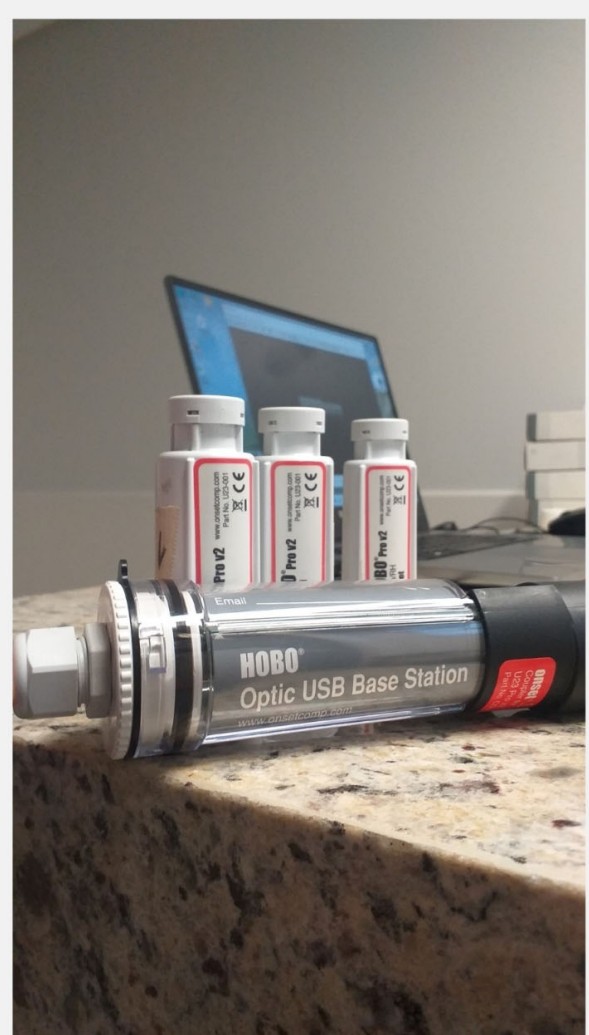

**Outdoor Temperature and Humidity Data Logger HOBO Pro V2 ONSET U23-001**

**Operating range:** - 40.0 °C to 70.0 °C
**Accuracy:** ± 0.21 °C (0,0 °C to 50.0 °C)
**Resolution:** 0.02 °C to 25.0 °C
**Response time:** 40 minutes in the air moving 1,0 m/s
**Deviation:** < 0,1 °C per year
**Operating range:** 0.0 % to 100.0 % relative humidity
**Accuracy:** ± 2.5 % from 10.0 % to 90.0 % relative humidity, to a maximum of 3.5 % including hysteresis at 25.0 ºC; Below 10.0 % and above 90.0 % ± 5.0 %.
**Resolution:** 0.05 %
**Dimensions:** 10.2 × 3,8 cm
**Response time:** 40 minutes in air moving 1.0 m/s with protective cap
**Deviation:** < 1.0 % per year
**Operating Temperature:** -40.0 °C to 70.0 °C
**Communication Type:** USB (U-4 Base Required)
**Recording Interval:** 1 second to 18 hours
**Btttery:** 1/2 AA, 3.6 volt lithium, user replaceable; 3-year battery life with logging interval of 1 minute or more.
**Memory:** 64KB (approximately 21,000 measurements)
**Protection degree:** NEMA 4
**Weight:** 57 g

**Figure 5.** HOBO® U23-001 model dataloggers used in mobile transects. Source: Authors (2022).

Regarding the MRT, it should be noted that it cannot be measured directly; thus, for its calculation, it was initially necessary to measure the globe temperature, which is a grey metal sphere (for open spaces with incidence of radiation) [46], with a measuring instrument at its center [20]. The TRM was calculated using Ta, Average Ws, and GT data [49,50], using the Equation (1) [43].

$$\text{MRT} = [(\text{Tg} + 273.15)4 + ((4.74 \times 107 \times va0.6) / (\varepsilon \times D0.4)) \times (\text{Tg} - \text{Ta})]1/4 = 273.15 \tag{1}$$

where:

MRT is the mean radiant temperature (°C);
GT is the globe temperature (°C);
$va$ is the air velocity at the level of the globe (m/s);
$\varepsilon$ is the emissivity of the globe (no dimension);
D is the diameter of the globe (m);
Ta is air temperature (°C);

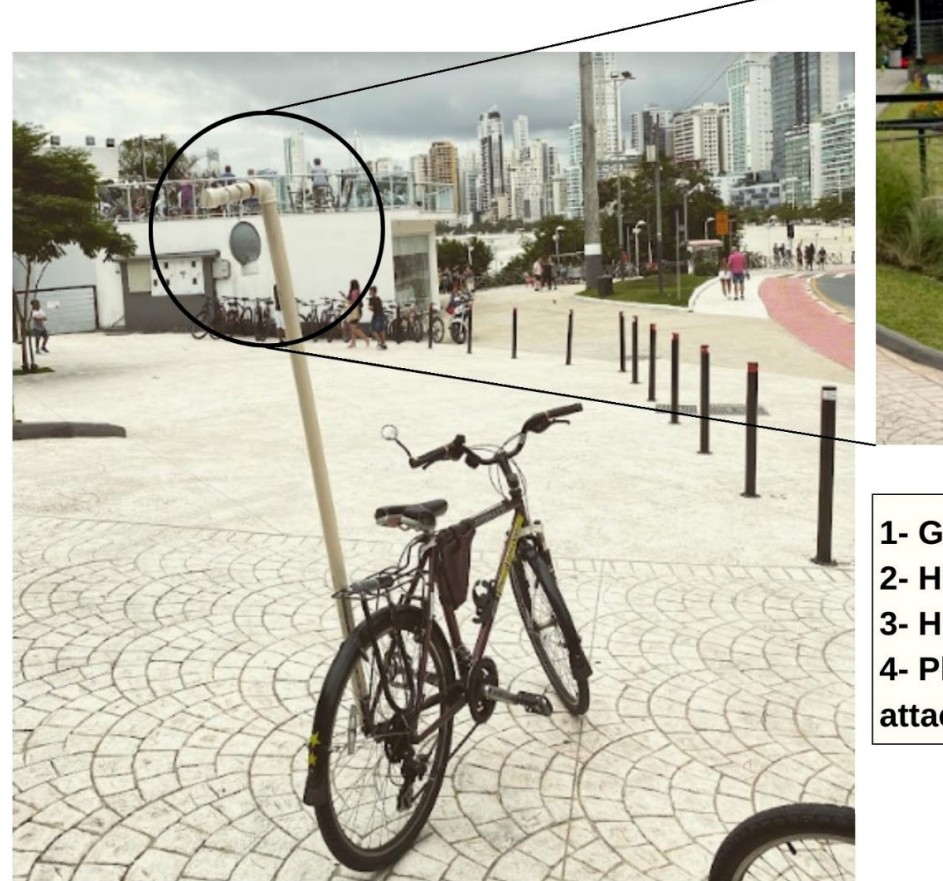
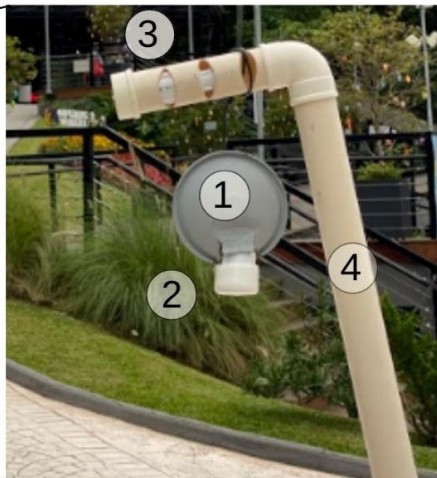

1- Grey globe for MRT
2- HOBO inside grey globe
3- HOBO for Ta and Rh
4- Plastic pipe frame, attached to the bicycle

**Figure 6.** Dataloggers attached to the bicycle. Source: Authors (2022).

Indexes of outdoor thermal comfort were used regarding environmental factors [48,49]. In order to carry out the measurements of Ta and air humidity, it was necessary to house the sensors to minimize the exchange of radiation [48]. The movement of air was measured by anemometers, accurate to variations in speed [20,48].

Regarding the MRT, it should be noted that it cannot be measured directly; thus, for its calculation, it was initially necessary to measure the Tg, which is a grey metal sphere (for open spaces with incidence of Sr), with a measuring instrument in its center [20] The TRM can be calculated through data of Ta, Ws, and Tg [50].

In addition, in order to increase the accuracy of the data collected, meteorological data obtained through 2 of the 24 fixed meteorological stations distributed throughout the city were used, which have been used in other research at the Laboratory of Climatology in Subtropical Environments (LaCAS) of the Federal University of Santa Maria (UFSM); these stations provided data on Sr and Ws [36].

### 2.4. Data Processing

After collecting data in the field, the information was processed and analyzed on the basis of the solar time of the location (UTC-3). Consequently, it was necessary to determine the location of the data provided by the dataloggers every 30 s [23]. Using variables provided by a free mobile sports application, each point in Universal Transverse Mercator (UTM) coordinates were determined by a straightforward calculation.

This enabled perfect knowledge of the overall distance, duration, and average speed of each transect. With this information, it was feasible to manually specialize the points within the Google Earth Pro 7.3 software, Free version for personal computer, using the marker and the ruler, among other tools. After processing the data, Qgis 3.0 Free version and

ArcGIS 10.5 software, licensed for the Department of Geosciences at the Federal University of Santa Maria. were used to generate maps. The maps that resulted from the research accounted for a distance of 75 m and a variety of environmental variables.

### 2.5. Outdoor Human Thermal Comfort Indices

Initially, environmental variables such as Ta, Rh, Ws, and Sr were connected to the RayMan PRO 2010 Free software (Freiburg, Germany) [42] in order to perform the necessary calculations. This program is utilised for the widest variety of objectives in urban-morphology-related investigations [41,42]. The Physiologically Equivalent Temperature (PET) [51] and the Universal Thermal Climate Index (UTCI) [18] were utilised for the analysis of human thermal comfort.

The physiological equivalent temperature (PET) describes the exposure environment in terms of the Ta that would be necessary under reference conditions (using the MEMI model) to produce the same thermal response [51]; additionally, from the adjustment for common clothing and activity characteristics for the thermal comfort assessment base, PET is very timely and universal in comparison with the majority of other indices, primarily due to the thermophysiological characteristics; Furthermore, it can be used in all seasons [41] because PET assumes 80 W activity and 0.9 clo for the reference environment. In addition to being a very relevant index, the UTCI provides accurate descriptions of body form, such as weight, height, age, gender, and internal transfers [52].

The interpretive ranges calibrated for the closest subtropical regions to the study site were utilised. A study was conducted in 2013 in Curitiba, Paraná, with the purpose of analyzing thermal comfort in open places in order to identify (dis)comfort ranges for the city using the UTCI index [18]. In 2018, the same procedure was used to determine the interpretive ranges for the city of Santa Maria (RS), the PET index, and a number of other variables [53,54].

### 3. Results

#### 3.1. Characteristics of Urban Morphology

The map of the LCZs available in the city of Balneário Camboriú was obtained through the WUDAPT [40]. With the images taken by the "fisheye" lens and later using the RayMan Pro program, it was possible to obtain the SVF calculations. Both results are available in Figure 7.

Figure 7 shows that, of the 17 existing LCZs [39], 10 are represented by the routes chosen to carry out the mobile measurements. It is possible to observe that the area closest to the ocean represents LCZ G and LCZ F and the highest concentration of LCZ 1 up to the Third Avenue (towards the continent). The western part of the city has mostly LCZ 3 concentration and also has significant areas of LCZ 8, and some A, B, and D. The city is practically divided between the two LCZs that stood out the most, LCZ 1 and LCZ 3.

It is noticed that the points with LCZ 1,with construction density, and LCZ 1 together with LCZ 4, which has the same height but different construction density, cause the SVF to decrease in these areas, comparatively more than the others [31]. LCZ 1 only has the lowest values of SVF, which are areas between $3^{rd}$ Avenue and Brasil Avenue (Points 3 and 11).

Point 5 and Point 12, which are on the edge and are receiving influences in areas with LCZ 1, LCZ G, and LCZ F, presented different SVF values, even though they are under the same area of influence. Point 5, unlike Point 12, is a little more wooded, and the buildings are taller. At Point 15, which is the last point of Route 2, LCZ 1 with LCZ 6 presented greater opening, due to the dense area of vegetation; areas with LCZ 3 also presented greater opening of SVF, and LCZ 3 together with LCZ 1 showed moderate SVF openings compared with the others. At Point 4, which is located on the riverbank, it was possible to perceive the large concentration of buildings in this area through the good opening of SFV.

LCZ 8, together with LCZ B, presented a large sky aperture at Point 7, as did LCZ 3 together with LCZ 6, at Point 1. LCZ 3 and LCZ 6 together vary the SVF aperture according to the layout of buildings and the distance between them [31].

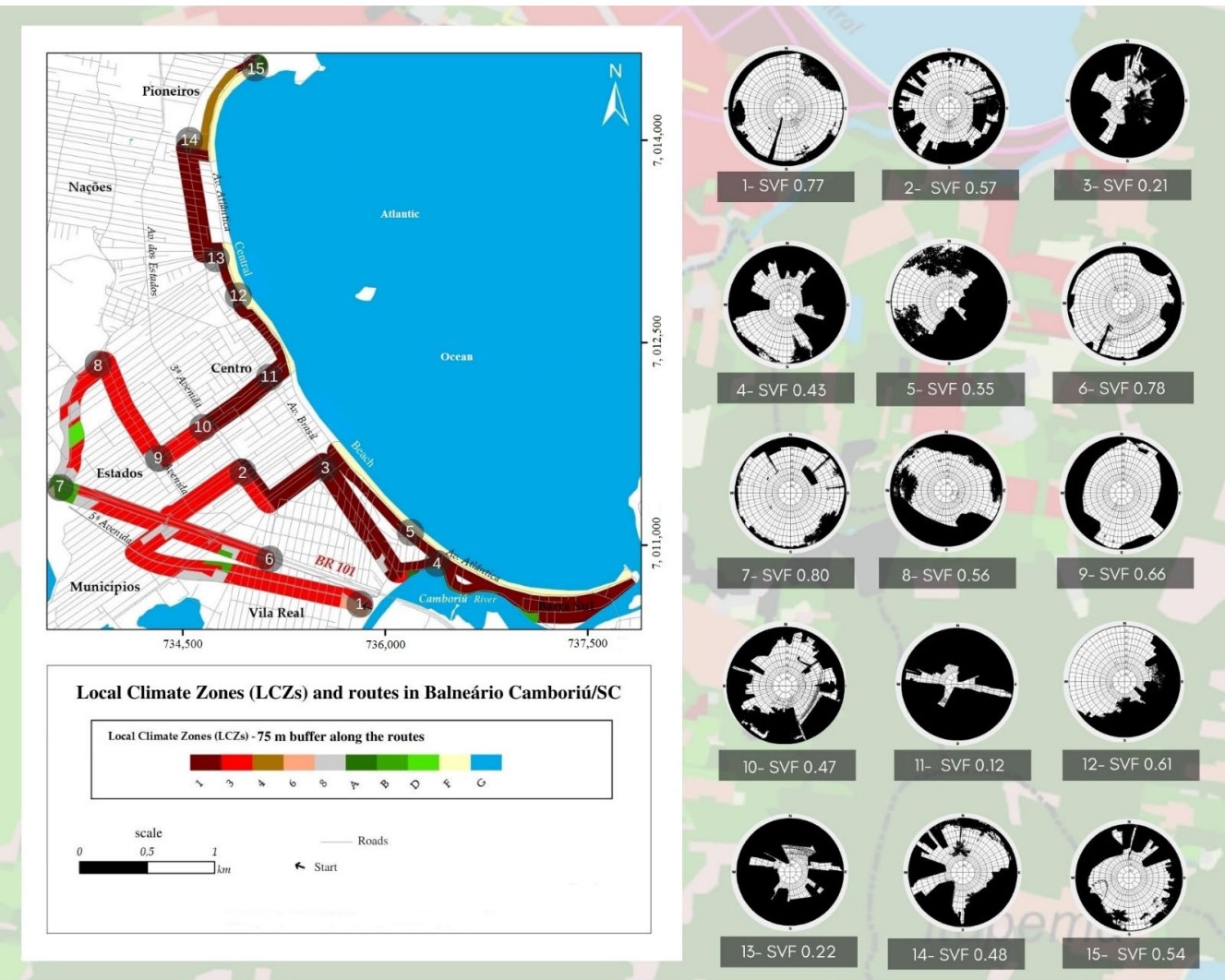

**Figure 7.** LCZs and SVFs available through transects. Source: Authors (2022).

*3.2. Environmental Variables Obtained through Mobile Measurements*

Figure 8 depicts the findings of Ta (°C), URA (%), and TRM (°C) readings for Route 1 and Route 2 at both times chosen for the mobile transects.

On 14 January 2022, at 9:00 a.m., the Ta (°C) ranged between 25.0 °C and 26.0 °C. The highest values were found mostly in the city's southern part, where Route 1 was located, at the following locations: The beginning of 5th Avenue, Corupá Street, the end of 2550 Street, across the 4th Avenue present in Route 1, a large portion of 3000 Street, and the intersection of 3000 Street and Brasil Avenue. In one section of Atlântica Avenue, there was a noticeable increase in value. On Route 2, the single place with the greatest Ta value at the time was on 2000 Street in the east, near to the ocean, heightened in comparison with those recorded at 9:00 a.m. It ranged from 26.0 °C to 29.0 °C at the time. Except for the commencement of the route, which had the highest value (29.0 °C), the lowest Ta values on Route 1 occurred on 5th Avenue; the remainder of the route had greater values from 2550 Street onward. The lowest Ta values on Route 2 were found on the Marginal Oeste Avenue, up to the Municípios Neighbourhood, and in the Barra Norte Neighbourhood (Atlântica and Brasil Avenues).

In terms of RH (%), the readings at 9:00 a.m. varied from 78.0% to 82.0% and remained consistent for the two routes across the city. It varied from 70.0% to 82.0% at 4:00 p.m. The Barra Norte and Barra Sul neighborhoods had the greatest values. The readings declined as one got closer to the water. The temperature on the MRT at 9:00 a.m. varied from 25.0 °C to

36.0 °C. The greatest values were recorded on Trip 1, mostly in Barra Norte and near the start of the route. The section of Route 1 with the lowest value was on 2550 Street, which had the city's lowest MRT value at 4:00 p.m.

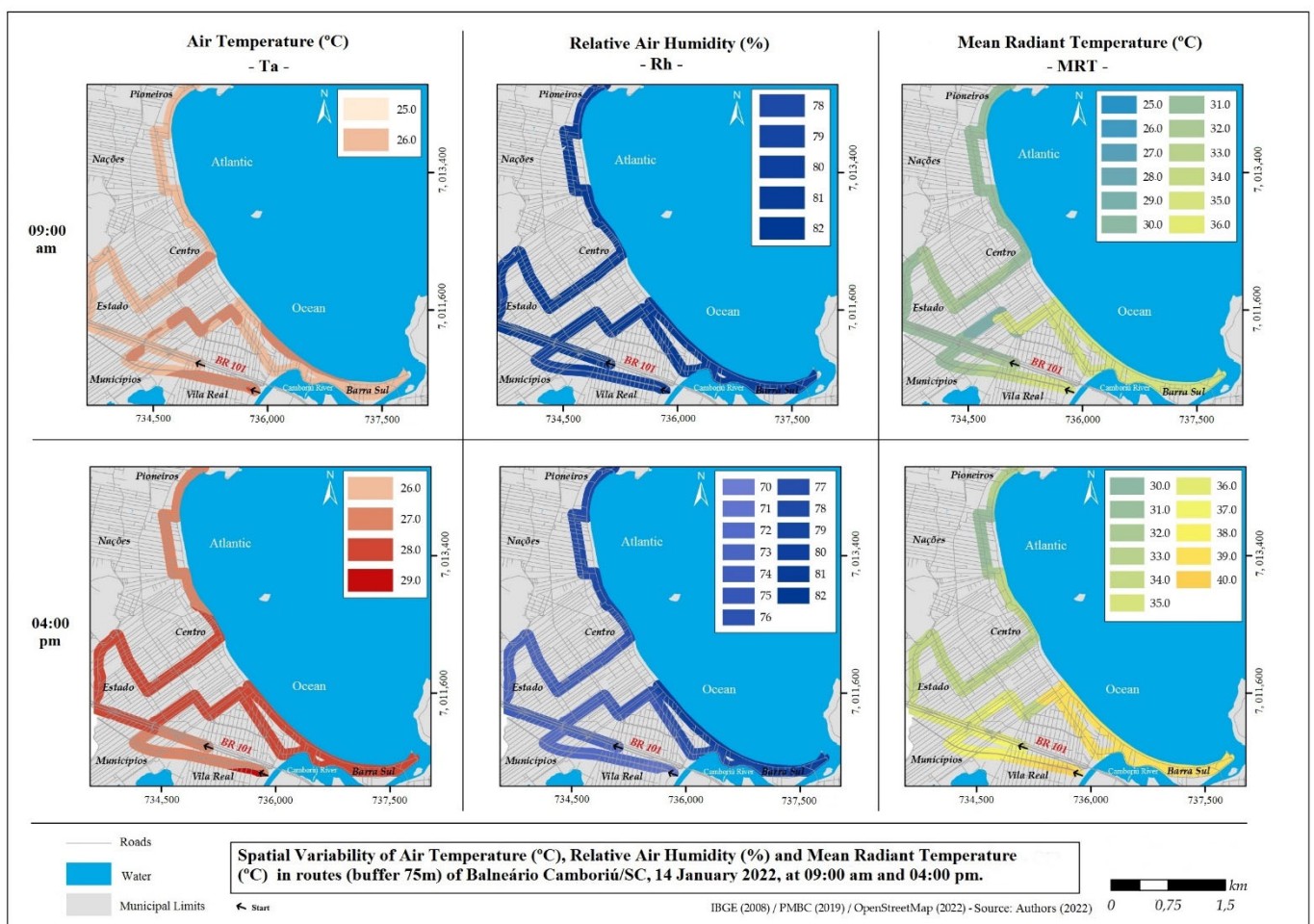

**Figure 8.** Spatial variability of Ta, RH, and MRT along Routes 1 and 2 (bike paths).

It was clearly seen that Route 2 had lower values than Route 1. We were able to observe that the MRT ranged from 30.0 °C to 40.0 °C at 4:00 p.m. The lowest values continued to occur along much of Route 2 (in the city's northern half) and the highest values in the city's southern part (Route 1). The beginning of Route 2, which is located on the city's southern outskirts, had the greatest value. The greatest value was found near the start of Route 1 on 5th Avenue, followed by Atlântica Avenue and Brasil Avenue.

### 3.3. Spatial Variability of Thermal (Dis)Comfort from the Indices

Through Figure 9, it is possible to observe the map produced from the data obtained for PET and UTCI for 14 January 2022, for the two routes at 9:00 a.m. and 4:00 p.m. Table 1 presents the ranges of (dis)comfort found through the PET and UTCI indexes, as well as the percentage (%) present in the routes (Table 1).

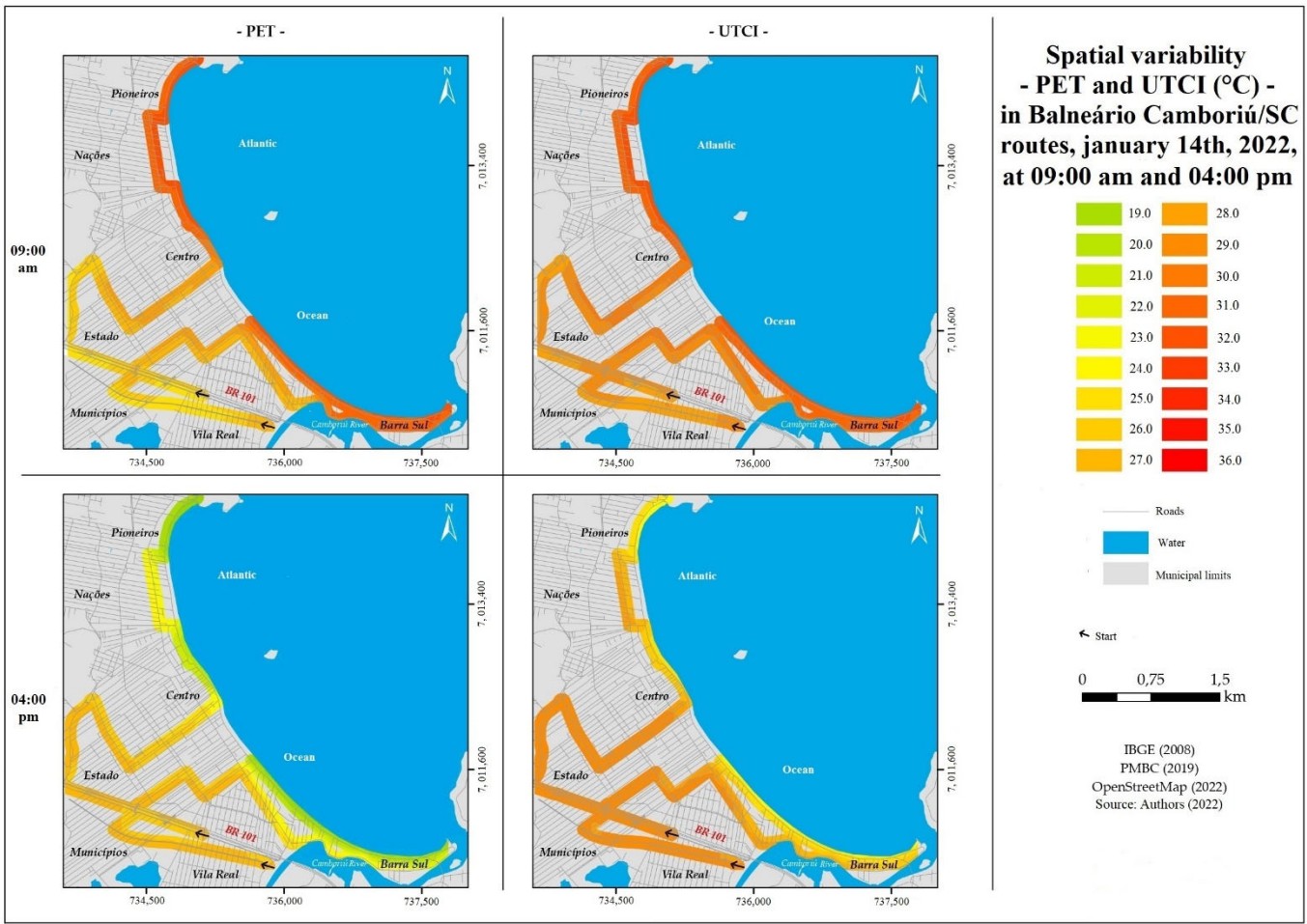

**Figure 9.** Spatial variability of PET and UTCI indices along bike paths in Balneário Camboriú/SC. Source: Authors (2022).

**Table 1.** Percentage of routes according to PET and UTCI interpretive ranges.

| Hour | PET | % of Routes | UTCI | % of Routes |
|---|---|---|---|---|
| 9:00 a.m. | "a little heat" | 60 | "comfortable" | 10 |
| | "heat" | 40 | "moderate heat stress" | 90 |
| 4:00 p.m. | "comfortable" | 24 | "comfortable" | 100 |
| | "heat" | 76 | | |

For the PET index at 9:00 a.m., it was possible to notice that according to the interpretative bands, all routes were uncomfortable, as the variation of the entire city was from 24.0 °C to 32.0 °C. According to the table, it is possible to observe that 60% of the route presented discomfort due to "a little heat" in the western part of the city; the points that presented higher values and fit the interpretative range of discomfort due to "heat" were 40% of the route, being Avenue Atlantic and Brasil Avenue on Route 1 and Streets 1001, 1901, and Brasil Avenue on Route 2. At 4:00 p.m., the city experienced discomfort in the western part of the city until Brasil Avenue and a small part of Barra Sul. According to the interpretive tracks, 76% of the journey was considered to have "discomfort due to a little heat". The lowest values were presented in 24% of the route, on Avenue Atlantic, where the city was comfortable.

For UTCI, at 9:00 a.m., most of the city showed discomfort. The points that showed comfort were 10% of the route: almost the entire 5th avenue on Route 1 and on Route 2, in

the initial part of the route, and some points between the Municipalities and Dos Estados neighborhoods. The highest values occurred in the eastern part of the city (Atlântica Avenue), in Barra Sul on Route 1 and on Brasil Avenue, 1001 Street, and 1901 Street in Route 2, which presented from 29.0 °C to 32.0 °C. The points of thermal discomfort were in the same interpretative range, which was "moderate heat stress" (90% of the route).

At 4:00 pm, for UTCI, the values ranged from 27.0 °C to 24.0 °C, making the city 100% comfortable at that time. In general, for both routes, the minimum values occurred at Atlântica Avenue, and the highest values occurred in the western part of the city, from Brasil Avenue. In Figure 10a,b, it is possible to observe the correlation between the SVF and the values (°C) found for PET and UTCI in the fifteen points available in Figures 7 and 8, at 9:00 a.m. and 4:00 p.m.

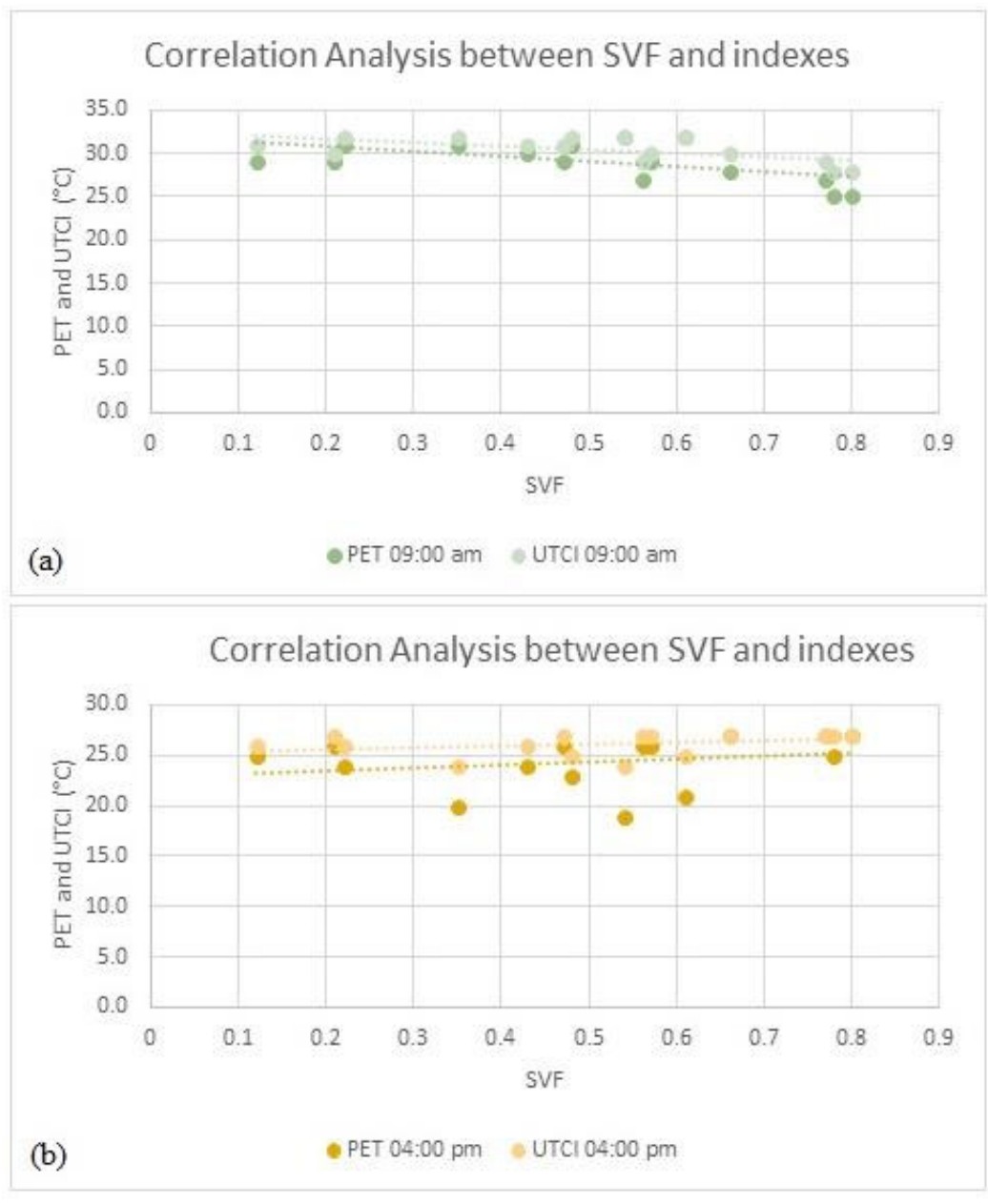

**Figure 10.** Correlation between SVF and indices at 9:00 a.m. (**a**) and 4:00 p.m. (**b**) along bike paths in Balneário Camboriú/SC. Source: Authors (2022).

Figure 10a,b show the linear correlation between the values of the PET and UTCI indices, on the Celsius (°C) scale, and the SVF calculations obtained at the 15 points observed

in Figure 8. In Figure 10, it is possible to observe that the values of the indexes represent the morning time (9:00 a.m.) and the analysis shows that, currently, there is a negative correlation between the SVF and the indexes; that is, the higher the SVF calculations, the lower the values are. Regarding values (°C) of the PET and UTCI indices in Figure 10, which shows the afternoon period, there is a positive correlation since the higher the SVF values, the higher the values of the PET and UTCI indices.

## 4. Discussion

### 4.1. Environmental Variables (Ta, RH, and MRT)

Ta was lower at 9:00 a.m. for both routes than at 4:00 p.m. This occurred because the regions had already stored more energy during the day at 4:00 p.m., as opposed to the time of the transects carried out at 9:00 a.m. [37]. The northernmost sector of the city (Barra Norte), represented by Route 2, had lower Ta readings during both eras. The coastline region, in contrast to the southern half (Barra Sul), has afforestation throughout the entire path, altering the Ta [3,7].

Points 13 and 14 (Figure 8) are in an area with buildings with dozens of stores and no permeability (Brasil Avenue, LCZ 1, low SVF), and will never have a positive radiation balance again due to the density of urban canyons [32,36,37]. Furthermore, as shown in field activities, the route that travels via the southern bar by the sea does not get additional solar radiation after 14:00 p.m., in contrast with the southern bar, which receives solar radiation for a longer period of time and, thus, retains a bigger quantity of energy [37].

Ta was higher near the start of Route 1 compared with the more continental parts of the routes (5th Avenue). This region is made up of LCZ 3 and has a large SVF aperture, enabling radiation to enter. The fact that it was the beginning of the transect and had been exposed to environmental conditions for some time prior to the transect's departure time may have impacted the temperature increase. The beginning of Route 2 (Marginal Oeste Avenue) would not have had this difficulty because it is a forested region with shadows, allowing the cyclist to wait until departure time without being exposed to the sun.

At 9:00 a.m., the temperature on Route 1 increased on Corupá Street. This is a tiny street in comparison with the others; it is crowded at this time of day, and it has a lot of concrete, which generates anthropogenic heat. It was also easy to detect that the end of the 2550 Street length had greater Ta. The quantity of movement of people and cars in this region increases, owing to trade and resulting in a rise in Ta; a similar situation occurred on 3rd Avenue, where LCZ 3 and LCZ 1 intersect with an SVF of 0.57.

In terms of relative humidity, there was a modest difference between the routes in the morning, with higher humidity than in the afternoon. It can be noted that in the afternoon, the RH stayed higher on Atântica Avenue and fell in the most continental section of the routes; it is known that this is connected to Ta (°C) [4], and, therefore, there are parallels between higher values of RH and lower Ta values, and vice versa. As a result, this is also connected to the peculiarities of urban morphology.

Route 1 (south of the city) has a higher MRT than Route 2 (north of the city). This is because the southern part of the city receives solar radiation for a longer period of time than the northern part, and, as a result, the material properties of the tall buildings present in this part of the city generate more heat to the environment, particularly those with glazed surfaces [36,37]. Furthermore, the low level of afforestation on the edge compared with Barra Norte may contribute to this, as growing trees lower the solar radiation absorption [3].

We were able to determine that the values acquired at 4:00 a.m. on both days were greater than those obtained in the morning. Brasil Avenue, located on Route 2, had the lowest values on all days and times. It is a street with a low SVF and LCZ 1; therefore, it is harmed by a lack of shortwave radiation [3,36,37].

### 4.2. Comfort Indexes and LCZ

The results indicate a pattern of thermal discomfort in the city based on the supplied schedules. At 9:00 a.m., there was a trend for a higher Ta of thermal discomfort in the coastal

section of the city, while at 4:00 p.m., the greatest discomfort is in the most continental section, with variations based on the indices, but exhibiting this same pattern. According to the interpretive ranges of the PET index for subtropical areas, discomfort was evident in the city at all times. With respect to the UTCI index ranges, comfort was present at 4:00 p.m.; however, the highest Ta remained on Brasil Avenue, the most continental part of the city [42–47].

On the basis of the data obtained from the fixed stations, it was possible to determine that the wind speeds at 9:00 a.m. were no more than 2 km/h in both trajectories, while at 4:00 p.m., they were 3 km/h on the mainland and between 6 km/h and 7 km/h on the coast, which contributed to the city feeling less uncomfortable at 4:00 p.m. that day.

The discomfort by the seaside in the morning may occur, in part, depending on the position of the city in relation to the sun, as the eastern part of the city already receives Sr for a longer period at that time, depending on the position of the buildings in relation to the sun. In addition, in this region, there is a high concentration of LCZ 1 with the lowest values of SVF, acting as a barrier to the most continental part of the city, which has LCZ 3. Greater values correspond to higher Ta (°C) values [31,33,39].

At 4:00 p.m., the opposite occurs near the sea, where the thermal comfort values are lowest, up to Brasil Avenue (LCZ 1). From there, towards the continent on both routes, it is possible to see values of discomfort; this is due to a number of causes. This is the section of the routes with LCZ 3, with buildings containing no more than three stories and a high SVF value; at this time, the quantity of energy stored by the city during the day is greater, resulting in a higher emission of long waves [36].

At 4:00 p.m., the edge which was being influenced by the wind, had the lowest numbers. The portions of Brasil Avenue present in the routes appeared to have higher values because of the absence of Sr due to the reduced SVF and the LCZ 1, which reduces its ventilation capacity. This causes anthropogenic heat not to be dissipated as it should in the MRT, despite the fact that this part of the city is highly populated by automobiles, bicycles, pedestrians, and other modes of transportation [3,6]. The north–northwest/south–southeast orientation streets do not receive adequate natural ventilation; therefore, the orientation of the urban fabric with respect to this discomfort [3] is also a factor.

Because the ventilation from the breeze is blocked by buildings, the orientation of the urban fabric in relation to the position of the sun influences this discomfort, resulting in streets with orientation north–northwest/south–southeast not receiving natural ventilation, which is essential for the mitigation of urban heat [3,51].

In addition, another process that occurs in relation to the low SVF and LCZ 1 is that as the shading process occurs during the day, the street canyon is in the shadow of the buildings, reducing the amount of Sr obtained by the canyon, causing the Ta to decrease as the SVF decreases; this may explain why at 4:00 p.m. there are values related to discomfort on Brasil Avenue on Route 2—even the Ta values present lower values in comparison with other parts of the routes; that is, they are lower than the average

When the SVF enhances heat dissipation in the street canyon, it can also allow the ground to obtain more Sr and cause the Ta to increase [3,6,36,55], as observed in LCZ 3 settings at approximately 4:00 p.m. on the mainland portion of the routes.

According to the results of both indices, it is probable that there will always be discomfort in the city between 9 and 10 a.m. on sunny summer days, typical of subtropical areas [6,53,54].

## 5. Conclusions

The level of human thermal (dis)comfort in connection with diverse surroundings on the bike paths of Balneário Camboriú in the summer varies substantially according to the interaction of the numerous parameters related to construction, position, and urban (im)permeability, as determined by this study. These interactions affect the environmental conditions that produce distinct microclimates. On both days shown, the amount of thermal discomfort appeared to be greater than that of comfort.

Utilizing the LCZs and SVF approach to characterize the urban morphology was effective for discussing the results, as the urban morphology has a significant impact on the external thermal environment. This research made it feasible to validate the approach of mobile transects for measuring environmental parameters in Balneário Camboriú via bicycle. According to the evidence presented, such an approach had never before been employed in Brazil, and it proved to be effective. In this regard, it is also important to note that creative approaches [56] that have never been undertaken in Brazil are crucial to the growth of climate geography research in the country, given the novelty and scarcity of this knowledge.

Regarding the limitations of the present study, it is anticipated that this methodology can be applied from the PET and UTCI interpretive ranges calibrated for Balneário Camboriú; thus, there may be a crossing of environmental variables with the personal factors of people acclimated to the regional climate. In addition, another relevant component that may be investigated in depth to contribute to the discussion of the results acquired from exterior human thermal (dis)comfort are the diagnoses pertaining to the building structure, such as the construction materials and building color.

Nonetheless, through the data obtained from mobile measurements in Balneário Camboriú, this work makes it possible to conduct different types of future analyses related to research involving external human thermal comfort and can provide funding to consider policies that aid in resolving issues related to thermal discomfort of people using bike paths to engage in physical activity [57,58].

**Author Contributions:** Conceptualization, C.A.W. and L.W.; methodology, C.A.W., L.W., J.P.A.G., I.T.C., S.S. and A.M.; software, C.A.W., L.W. and A.M.; validation, L.W., C.A.W., J.P.A.G., S.S. and A.M.; formal analysis, A.M. and S.S.; investigation, L.W. and C.A.W.; resources, C.A.W. and J.P.A.G.; data curation, C.A.W., L.W. and I.T.C.; writing—original draft preparation, C.A.W. and L.W.; writing—review and editing, C.A.W., L.W., A.M. and S.S.; visualization, C.A.W., J.P.A.G. and S.S.; supervision, C.A.W., J.P.A.G., S.S. and A.M.; project administration, C.A.W.; funding acquisition, C.A.W. All authors have read and agreed to the published version of the manuscript.

**Funding:** This study was financed in part by the Coordenação de Aperfeiçoamento de Pessoal de Nível Superior-Brasil (CAPES)-Finance Code 001. Conselho Nacional de Desenvolvimento Científico e Tecnológico (CNPq) for proving the Research and Productivity research: grant process number 306505/2020-7.

**Informed Consent Statement:** Not applicable.

**Acknowledgments:** We thank the Conselho Nacional de Desenvolvimento Científico e Tecnológico (CNPq) for proving the Research and Productivity research: grant process number 306505/2020-7.

**Conflicts of Interest:** The authors declare no conflict of interest.

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
