# Peer review of "Outdoor Human Thermal Comfort along Bike Paths in Balneário Camboriú/SC, Brazil"

_atmosphere, doi:10.3390/atmos13122092_

Round 1

Reviewer 1 Report

The submitted manuscript entitled “Outdoor human thermal comfort along bike paths in Balneário 2 Camboriú/SC, Brazil” implements the characterization of the urban morphology of the study area using the Local Climatic Zones and the Sky Vision Factor, to contribute to the choice of routes to be traveled by the existing cycle routes in the city. The research is addressing a valid research gap, concerning the method, the application of SVF and LCZ to quantify the outdoor thermal comfort has not been fully investigated, however the manuscript needs major revisions before being acceptable to be published.

·       The study needs to add relevant literature regarding the outdoor comfort and bike riding in cities. Examples are: https://doi.org/10.1016/j.buildenv.2021.108577

·       It is not clear how the experiment collects the wind speed. There are no sensors mentioned in the experiment setup to collect the local wind on the bikers. In case the study is using rooftop stations, how does that translate to the local 2m height wind speed on the street?

·       Please Include a map with the MRT variations on the routes. That would be more interesting since MRT a spatio-temporal parameter incontrast to SVF and LCZ which are both static parameters

·       How does the research consider the accumulation effects of heat on a path? On a path how does the variation of microclimate conditions correspond to the final equivalent temperature experienced by the person?

·       One interesting finding of the research is that the frequencies of comfort and discomfort is different for PET and UTCI, this needs to be investigated further and see which one is more reliable for this context / climate / application case.

·       For further studies the methodology can also use the geo-referenced method to collect the geo-tagged microclimate data: https://doi.org/10.1016/j.jobe.2018.07.003

Author Response

Answers attached.

Author Response

Answers attached.

Reviewer 3 Report

This study evaluated the outdoor human thermal comfort level along bike paths in Balneario Camborius, Brazi in a specific data (January 14th 2022) using two morphology indices including Local Climatic Zones and the Sky Vision Factor, and two thermal comfort indices including PET and UTCI. Seem like the current manuscript presented some knowledge, however, as a reviewer, the only concern came from the insufficient contributions. This study can only be a case report instead of a deep research. The detailed comments can be found below. 

(1) The sensors in the mobile measurement should be clearly described including the accuracy, resolution, range.

(2) This study considered two mobile measurement period, which are 9:00 am, and 4:00pm. Reviewer did not wonder its reasonable decision, however, reviewer wants to know how long one route took when the experiment were conducted. In other words, whether the two cases, regarding the descriptions of “9:00 am and 4:00pm” were reasonable.

(3) What are the local weather condition, 9:00 am and 4:00pm on January 14th 2022? This info should be provided in a detailed way.

(4) The correlations between SVF and PET/UTCI were not sufficient. The current data cannot present the correlation well. Please provide more data.

(5) As a cases study, the current manuscript only presented two routes and gave the conclusion, e.g., the perceptual of routes according to the PET and UTCI interpretive ranges, (The PET index showed that 24% of the routes were comfortable), which are not acceptable.

(6) This study did not present sufficient contribution, also the novelty of this article was not presented well.

(7) The purpose of this study, “to characterize the urban morphology of the research area, validate mobile measurements with bicycles in Brazil, and assess the thermal (dis)comfort of humans” were not clearly written.

Author Response

Answers attached.

Round 2

Reviewer 1 Report

Thank you for considering the suggestions and further improving the manuscript.

Author Response

Dear reviewer.

Thank you very much for all the placements and suggestions at work. They were really important for the enrichment of the research, and we tried to be attentive to what was requested. We are very grateful for your opinion.

Authors.

Reviewer 3 Report

The current manuscript can be accepted after a series of modifications. 

Author Response

Thank you very much for all the placements and suggestions at work. They were really important for the enrichment of the research, and we tried to be attentive to what was requested. We are very grateful for your opinion.